# Molecular Simulation of Methane Adsorption Capacity of Matrix Components of Shale

**DOI:** 10.3390/nano12224037

**Published:** 2022-11-17

**Authors:** Xiaoxue Liu, Zhenxue Jiang, Shibin Liu, Bo Zhang, Kun Zhang, Xianglu Tang

**Affiliations:** 1State Key Laboratory of Petroleum Resources and Prospecting, China University of Petroleum, Beijing 102249, China; 2Unconventional Oil and Gas Science Technology Research Institute, China University of Petroleum (Beijing), Beijing 102249, China; 3Gudong Oil Production Plant, Sinopec Shengli Oilfield Co., Ltd., Dongying 257000, China; 4Schlumberger Houston Production Technology Center, Houston, TX 77042, USA; 5School of Geoscience and Technology, Southwest Petroleum University, Chengdu 610000, China

**Keywords:** molecular simulation, methane adsorption capacity, matrix components, marine shale

## Abstract

Shale gas occurs mainly as adsorption and free gas. Among them, whether the adsorbed gas can be gradually desorbed or not is a major cause of stable and high yield. The matrix component is the main factor affecting the adsorption capacity of shale. In this paper, by simulation software named Materials Studio (MS), using Molecular Dynamics Simulation and Monte Carlo Simulation, the adsorption capacity of different matrix components under specific conditions is studied and the four models: relative concentration model, diffusion coefficient model, saturated adsorption capacity model and isosteric heat of adsorption model, are built. The simulation models show that the mineral matrix has a significant impact on the adsorption of methane molecules in shale: kerogen I > smectite > chlorite > illite > quartz. Kerogen I has the strongest adsorption capacity with high-density thick layer adsorption. Under the temperature (369.97 K) and the formation pressure (28.07 MPa) and under the condition of 6.0 nm in the cylindrical hole, excess adsorption amount of kerogen I is 13.418%, the diffusion coefficient is only 0.046 Å^2^/ps, saturated adsorption amount is 3.060 cm^3^/g, and the amount of adsorption heat is 9.598 kJ/mol. As the adsorption force on the pore wall is not as strong as the interaction repulsion force between adsorbents within a short distance, the clay minerals all have 2~4 layers of narrow layer and low-density adsorption. The adsorption thickness of the single layer is inversely proportional to its adsorption capacity, and the adsorption capacity is positively correlated with the opportunity of exposing oxygen atoms to form hydrogen bonds. Quartz has no obvious adsorption potential for methane molecules. This study is conducive to the quantitative evaluation of shale gas adsorption capacity, selection of favorable blocks and advantageous zones of shale gas reservoirs, and the improvement of development efficiency.

## 1. Introduction

Shale gas is a closed unconventional resource that self-generates and self-accumulates in dark shale or high-carbon shale, occurring as adsorbed gas, free gas and solution gas [1,2,3]. Among them, the first two are the main form of shale gas. Adsorption is one of the basic mechanisms for the accumulation of shale gas. According to the developed shale gas in the United States, the content of adsorbed gas accounts for 20~85% of the total shale gas content [1]. The production life of shale gas well is generally long and the rate of decline in annual production is lower than 5%, as it is able to retain a stable yield for dozens of years. In the present study, this was found to be closely related to the content of adsorbed gas. In the process of exploitation, free gas is firstly produced, and then adsorbed gas is desorbed gradually with a pressure decrease. Whether or not the adsorbed gas can be gradually desorbed is a major cause of stable and high yield [4].

Many previous studies on the adsorption capacity of shale have shown that there are many factors affecting the adsorption capacity of methane. Different matrix components have different media interface effects, leading to variation in the adsorption capacity of shale gas [5,6]. Predecessors have found that compared with inorganic components, organic components contribute more to the adsorption capacity. Adsorbed gas mainly exists on microporous and mesoporous surfaces, and organic matter can provide more specific surface areas for gas adsorption [7,8,9]. At the same time, the adsorption heat of organic components is larger than that of inorganic components, and the affinity of molecular groups to methane is strong, so organic components make more of a contribution to the adsorption capacity [10]. Zhang et al. [11] proposed that different types of kerogen has different adsorption capacities of methane, and the order of adsorption capacities was found to be: type III > type II > type I. Their study also pointed out that the affinity of aromatic kerogen to methane was higher than that of kerogen containing more fatty organic matter. For different inorganic minerals, the adsorption capacity of methane is significantly different. Clay minerals have rich micropores, which provide a large amount of specific surface area for adsorption. However, brittle minerals, such as quartz and calcite in shale, are weak in absorbing methane due to their small specific surface area [6,12]. Different clay minerals have different pore volumes and surface areas, resulting in the different adsorption capacity of methane [13,14]. Montgomery et al. [15], Loucks and Ruppel [16] and Hao et al. [12] believed that pore structure was an important factor affecting shale adsorption capacity, and gas adsorption was mainly related to micropores. In the Paleozoic marine organic-rich shale reservoirs with a high degree of evolution, there are a large number of organic matter pores (the diameters of pores are usually less than 50 nm). The free space of pores is smaller, and the specific surface area is larger, and the proportion of adsorbed gas is higher. However, the development of organic matter pores in low-abundance marine and terrestrial shale is limited, and the reservoir space of shale is mainly composed of inorganic pores with pore sizes ranging 30~4.5 um, which have a lower adsorption capacity than organic matter pores and the proportion of adsorbed gas is lower. Shale gas adsorption dominated by physical adsorption is an exothermic process. High temperature can inhibit exothermic adsorption and reduce the adsorption of gas, and the content of adsorbed decreases with the increase in temperature [17,18,19,20]. Under conditions of low pressure, the content of adsorbed gas increases slowly with the increase in pressure and reaches the maximum value. Under conditions of high pressure, due to the destruction of the sample structure and the change in intermolecular force, the content of absorbed gas shows a slow linear decline trend with the increase in pressure. In addition, the buried depth, uplift and denudation of the formation, with changes in temperature and pressure, also affect the interface effects of the media, leading to the transformation of the modes of occurrence of shale gas [20].

Currently, research on adsorbed gas is generally based on macroscopic physical simulation experiments such as methane isothermal adsorption experiments. The applied interpretation models are all based on monolayer adsorption of adsorbents, which can only measure the adsorption capacity by saturated adsorbed gas, but cannot quantitatively evaluate the adsorption capacity of shale gas. Traditional normal temperature and pressure models, such as the Langmuir model, are used to explain the layered adsorption of supercritical methane under high temperature and pressure, but lack discussion of adsorption mechanisms in underground burial conditions. Although some scholars have used the molecular simulation method to study the adsorption performance of shale reservoir to shale gas, the pore models established are mostly simple slit pores or carbon nanopores, which have single structure and single molecular properties on the exposed matrix surface [21,22,23,24]. The modeling of micro and nano pores is not closely combined with the actual geological conditions. Therefore, based on the actual geological conditions (formation temperature, formation pressure, mineral composition, pore structure), cylindrical pores perpendicular to the short side of the matrix were selected as the pore form in this paper, which can not only fit the circular pore shape well in the XY plane, but also simulate the slit shape in the Z direction of the vertical plane. Molecular Dynamics Simulation and Monte Carlo Simulation are used for the adsorption simulation experiment in nanometer pores of different matrix compositions. The distribution characteristic of supercritical adsorption of methane in shale is determined, and occurrence state models and the quantitative evaluation models are built. The Polanyi adsorption potential theory [25] was used to explain the influence of different adsorption conditions on the adsorption content and distribution of methane in nanometer pores, which is conducive to the selection of favorable blocks and advantageous strata of shale gas reservoirs and the improvement of exploitation efficiency.

### 1.1. Parameters of Simulation

According to the geological features of marine shale in southern China, current formation temperature (369.97 K) and the formation pressure (28.07 MPa) were selected as conditions of shale. Simulation force field was COMPASSII force field, an ab initio force-field optimized for condensed-phase applications. The all-atom model was adopted. 60 Å was selected as the pore diameter and cylindrical pore perpendicular to the short edge of the matrix was selected as the pore shape, which could not only fit the circular pore shape well on the XY plane, but also simulate the slit shape on the Z direction of the vertical plane (Figure 1). Furthermore, organic matter (kerogen I), inorganic clay mineral (smectite, chlorite and illite), and inorganic siliceous mineral (quartz) were selected as the interfacial matrix for simulation, namely adsorbent. The sorbent single crystal cells were obtained from the ICSD2009 crystal structure database. The chemical formula of kerogen I is (C_251_H_385_O_13_N_7_S_3_), smectite is Al_2_[Si_4_O_10_](OH)_2_, chlorite is Mg_4_FeAl[AlSi_3_O_10_](OH)_8_, illite is KAl_2_[(OH)_2_AlSi_3_O_10_] and quartz is SiO_2_ [26,27,28,29,30]. For each inorganic mineral and kerogen cell in three spatial directions (the origin is O, and the three sides are OA, OB, OC in turn), the matrix with the size of 80 Å × 80 Å × 30 Å was made by the supercell expansion tool (Table 1). Since the C plane angle of chlorite and quartz single crystal cell is not a right angle, it is necessary to convert the side length when calculating the projection tetragonal system.

### 1.2. Method of Simulation

In this paper, the Gradient Descent was adopted to finish geometry optimization of the energy of the absorption configuration and the location of minimum energy point on the potential energy surface was determined. This method conducted a linear search along the gradient direction of energy descent. When the energy converged and approached the minimum point of energy, the search direction was oscillated and the convergence speed decreased. Finally, the low-energy structure of the system was determined, and the stable structure of the configuration was obtained.

Monte Carlo Simulation and Molecular Dynamics Simulation were used.

The specific steps of Monte Carlo Simulation were as follows. Firstly, isothermal adsorption simulation and isobaric adsorption simulation were, respectively, carried out for molecular pore model after geometry optimization. In the isothermal adsorption simulation, the type, starting and target fugacity of adsorbate were set, and in the isobar bed adsorption simulation, the type and target fugacity of adsorbate were set. Secondly, using the method of Metropolis [31] to carry on the importance of sampling, Compass II potential energy function was chosen, and the truncation radius of Van Der Waals force (12.5 Å), balance step number (10,000) and production step number (20,000) were specified for Monte Carlo adsorption simulation. Finally, in each experiment, 11 slices were taken from the iteration as step test points, and relevant physical quantities under the current configuration were counted.

The specific steps of Molecular Dynamics Simulation were as follows. Firstly, for molecular pore model after geometry optimization, the NVE system was adopted to anneal and search the optimal solution. Compass II potential energy function was used, and the truncation radius of Van Der Waals force was specified as 12.5 Å. After simulation of three thermal cycling of 10,500 steps, a stable state was achieved, reducing the difficulty of the simulation. Secondly, by the NVT system and Compass II potential function, a dynamic simulation was conducted under conditions of step size of 1 fs, a total of 10^6^ steps, total time of 1000 ps, until equilibrium. Finally, in each experiment, 1 ps was taken as the statistical point to count the relevant physical quantities in the current simulated state.

## 2. Results and Discussion

### 2.1. Relative Density

The relative density is the concentration profile of the three-dimensional periodic structure arranged in the one-dimensional radial direction through the statistics of the number of molecules in each parallel plane in the three-dimensional simulation system [32]. It is equivalent to taking the three-dimensional coordinate component of each molecule, projecting the molecule onto the corresponding component and accumulating its number. Based on the area integral of the difference in relative density between the actual distribution and the uniform distribution, the adsorption amount is added up and the proportion of adsorbed gas in the total gas content is determined. The relative density peak derived from the maximum relative density can be used to judge the strength of adsorption stratification of the nano pores of shale.

The relative density of the real uniform distribution should be obtained through the conversion of Formula (1):(1)$$C_{\mathrm{avg}}=\frac{V_{\mathrm{all}}}{V_{\mathrm{pore}}}$$
where *C*_avg_ refers to the average relative density under uniform distribution; *V*_all_ refers to the total volume of system, Å^3^; *V*_pore_ refers to the volume of the pore space, Å^3^.

By form of relative density, kerogen I with the best adsorption capacity shows the significant single layer adsorption, with large thickness and high peak, whose average thickness is up to 7.750 Å and relative density is up to 2.092. The degree of aggregation is very high, and the corresponding density of the molecular adsorption layer is also the biggest, significantly prior to the other mineral matrix. In addition, the absorption peak is significantly close to the wall of pore, with the largest displacement of adsorption. The smectite, with the second-best adsorption capacity, is an obvious adsorption of three layers, and the adsorption thickness of each layer is very low. The total adsorption layer thickness is 12.660 Å, and the average adsorption layer is 4.220 Å. Furthermore, the density of the second and third layer are higher, with a relative density of 1.556. The chlorite, whose adsorption capacity is worse than smectite, is shown as 3~4 layers of narrow layer adsorption, and the potential well between adsorption layers is relatively low, with no obvious distinction. The total thickness of adsorption layers is 17.375 Å, with an average of 4.344 Å, and the peak of relative density is 1.386 (Figure 2). Moreover, the illite, with worse absorption capacity than chlorite, shows the 2~3 layers of layer adsorption and potential well between the adsorption layers is relatively high, which results in an obvious low-density interval between the adsorption layers. The total thickness of adsorption layers is 14.750 Å, with an average of 4.917 Å, and the peak of relative density is 1.631. Quartz does not show obvious distribution characteristics of adsorption, and the peak of relative density is 1.372.

Through the calculation of the relative density, the proportions of the excess adsorption amount of several common minerals under the pore diameter of 60 Å diameter, formation temperature of 369.97 K and pressure of 28.07 MPa are obtained, as shown in the Figure 3.

The results obtained by the five groups of experiments are basically consistent with the qualitative results obtained by previous isothermal adsorption physical simulation experiments. The peak value of relative density of kerogen is the highest (2.092), and that of quartz is the lowest. For clay minerals, smectite and chlorite have more adsorption layers and less molecular density of methane in monolayer adsorption, so the peak value of relative density is lower than that of illite. Because there is a great quantity of hydrogen and oxygen atoms in kerogen I, they can combine hydrogen atoms of methane to form a large amount of hydrogen bonds. Moreover, the stronger the adsorption force is, the more adsorbents can be absorbed in the unit space, and the more methane molecules with high adsorption displacement. Adsorption layer thickness of kerogen I is large, considering the methane molecule diameter of 4.14 Å [33,34,35], so there is at least a two-layer thickness of high-density methane molecular aggregation. This is because the wall had strong adsorption force, which is stronger than mutual repulsion force between the adsorbate molecules. So, methane absorption in kerogen I is unable to show good stratification, is rather dense together (Figure 4a), and therefore kerogen I has the largest adsorption capacity.

Methane absorption in clay minerals has a multilayer, and the thickness of each layer adsorption is thin, about 4~5 Å. Adsorption in clay minerals has shallow potential well, and each layer has low density and a relatively orderly adsorption distribution. This is because the Van Der Waals force is the main force between the adsorbent and adsorbate molecules for clay minerals content, which is relatively weaker than the hydrogen bond. As a result, when methane molecules are closer to each other, to a certain extent, the molecular repulsive force between them rapidly rises according to LJ potential function (Figure 4b), overcoming the adsorption potential. So, a layer cannot be filled, and the lower-density multiple adsorption layers occur in the aggregation layer. Moreover, the thickness of the monolayer adsorption layer is inversely proportional to the strength of the adsorption potential. That is to say, the stronger the adsorption potential is, the smaller the intermolecular distance is, and the lower the monolayer thickness is. For example, the proportion of excess adsorption amount of smectite is greater than that of chlorite and that of chlorite is greater than that of illite, and the corresponding adsorption layer thickness is 4.22Å, less than that of chlorite and that of chlorite is 4.34 Å, less than that of illite (4.92 Å). Due to the low potential energy and high kinetic energy, the adsorption layer near the center of the pore wall is more prone to desorption than the adsorption layer near the pore wall, having the characteristics of molecular thermal motion under state of high temperature. Relative to the kerogen I, methane in clay minerals is more conducive to development. Quartz, as an inorganic silica mineral whose surface group is relatively single, belongs to the polar molecules. The forces between polar adsorbent and methane (nonpolar molecule) are mainly an induction force and the dispersion force, without orientation force. The silicon atoms exposed are stable, lacking hydrogen bonds, which is far stronger than the Van Der Waals force. So, the adsorption potential is low and almost no displacement of adsorption from the form of the distribution pattern occurs.

### 2.2. Diffusion Coefficient

Diffusion coefficient is an important index to measure the motion intensity of microscopic substances. The larger the diffusion coefficient is, the higher the spatial freedom of adsorbed plasma molecules is, and the weaker the adsorption effect is, which reflects the extent of binding degree of gas adsorbed molecules after being adsorbed by a different matrix [36,37,38].

According to the Einstein equation, the diffusion coefficient of CH_4_ in the pores is calculated based on mean square displacement of matter per unit time. This diffusion coefficient can be derived through non-equilibrium statistical thermodynamics, as shown in Formula (2).
(2)$$D=\underset{t\to \infty }{\lim}\frac{1}{6t}\left(\frac{1}{N_{t}}\sum_{i=1}^{N}{\left|r_{i}(t)-r_{i}(0)\right|}^{2}\right)$$

Mean square displacement can be calculated by Formula (3).
(3)$$MSD={\left|r_{i}(t)-r_{i}(0)\right|}^{2}=\frac{1}{NN_{t}}\sum_{i=1}^{N}\cdot \sum_{i=1}^{N}{\left|r_{i}\left(t+t_{0}\right)-r_{i}\left(t_{0}\right)\right|}^{2}=\underset{t\to \infty }{\lim}\left(\frac{1}{N_{t}}\sum_{i=1}^{N}{\left|r_{i}(t)-r_{i}(0)\right|}^{2}\right)$$

Thus, the linear slope (*K_msd_*) of mean square displacement is expressed as the Formula (4).
(4)$$K_{msd}=\underset{t\to \infty }{\lim}\frac{1}{t}\left(\frac{1}{N_{t}}\sum_{i=1}^{N}{\left|r_{i}(t)-r_{i}(0)\right|}^{2}\right)$$
where *t* refers to the simulation time, ps; *N* refers to the total number of adsorbent molecules; *N_t_* refers to the statistical average number of molecular dynamics steps; *r_i_*(*t*) refers to the true displacement of the *i*-th particle’s center of mass at *t* time.

By combining Formulas (2) and (4), the slope of the mean square displacement in the three-dimensional space is 6 times that of the diffusion coefficient. In other words, if the slope of the linear relation between the mean square displacement and the simulation time is taken as 1/6, the diffusion coefficient of methane molecules in the pores corresponding to different minerals can be obtained.

According to Figure 5, under the current formation temperature (369.97 K) and formation pressure (28.07 MPa), the diffusion coefficient shows the following trend when the water saturation is 0% and diameters of pores are 60Å:

Kerogen I < smectite < chlorite < illite < quartz

Previous studies on methane diffusion of single mineral for these materials show that under the same temperature and pressure, the diffusion coefficient of methane is quartz > illite > chlorite > montmorillonite > kerogen [39], which is consistent with the simulation result in this paper.

In XY direction and Z direction, the same conclusion is shown, which is basically consistent with the conclusion of relative density. Diffusion coefficient of methane molecular of kerogen I is the smallest one and lots of methane molecules are bound on the pore wall and the degree of freedom is low. Because many hydrogen bonds between kerogen I and methane molecules, and high adsorption potential, most kinetic energy of methane molecules can be converted to the Van Der Waals potential energy and hydrogen bonds binding energy, so methane can be attached near the pore wall. In addition, the adsorption site produced by the combination of an oxygen atom and a hydrogen atom in clay minerals make clay minerals have a certain adsorption potential. As a result, the diffusion coefficient of clay minerals is higher than kerogen but lower than quartz. Furthermore, the methane molecule in quartz has the greatest diffusion coefficient and the weakest adsorption effect.

### 2.3. Saturated Adsorption Amount Model

Saturated adsorption amount reflects the upper limit of adsorption capacity of corresponding pores [40,41]. By transforming the Langmuir equation, the number of adsorbed molecules and the intercept of the linear relationship between the amounts of adsorbed molecules divided by the fugacity are calculated. Then, the number of saturated adsorbed molecules in the pores of the corresponding matrix can be calculated, and the concentration of gas content can be converted. Finally, the saturated adsorption amounts are obtained (Formula (5)). Results are as shown in Figure 6.
(5)$$B=\frac{A}{V_{\mathrm{free}}N_{A}}\cdot \phi /\rho \cdot 1000\cdot V_{m}$$

In the above, *A* refers to the number of adsorbed molecules in the system; *B* refers to the adsorption amounts, cm^3^/g; *ρ* refers to sample density, g/m^3^; φ refers to the average porosity of the sample, %; *N_A_* refers Avogadro’s number; *V_m_* refers the molar volume of gas, which is 22.4 L in the ideal state. *V*_free_ refers to the free volume in the pore, Å^3^, which can be calculated by the modeling tool Atom Volumns & Surfaces, as shown in Table 2.

For adsorption capacity, descending order is Kerogen I, smectite, chlorite, illite and quartz. The adsorption capacity of Kerogen I is the strongest and saturated adsorption amount is 3.060 cm^3^/g, and adsorption capacity of quartz is the worst. Single mineral methane isothermal adsorption experiments have been carried out, and the experimental results show that the adsorption capacity is from strong to weak: kerogen > montmorillonite > chlorite > illite > quartz [42,43], which is consistent with the simulation result in this paper.

Moreover, the mineral composition can be obtained through XRD experiment, and it is assumed that the simulated saturated adsorption amounts of the feldspar, pyrite and other minerals are similar to quartz. Then, the saturated adsorption amount can be calculated by Formula (6).
(6)$$C_{\mathrm{all}\;}=\sum C_{\mathrm{mineral}\;}P_{\mathrm{minetral}\;}$$
where *C*_mineral_ refers to simulated saturated adsorption amount, cm^3^/g, and *P*_mineral_ refers to proportions of different minerals, %.

For example, according to the XRD results of marine shale from Southern China, the matrix is dominated by quartz and supplemented by clay minerals. Quartz averagely accounts for 57.34%, and clay minerals account for 24.55%. Among the clay minerals, illite is the most developed, accounting for 15.55%, followed by smecitite, accounting for 7.11% and chlorite only accounts for 1.89%. Furthermore, total organic matter is 7.45%. According to Formula (1), saturated adsorption amount is 2.55 cm^3^/g, close to average saturation adsorption amount of isothermal adsorption experiment (2.38 cm^3^/g), which proves that the Monte Carlo isothermal adsorption simulation model is close to real geological conditions and has evaluation significance.

### 2.4. Isosteric Heat of Adsorption Model

Isosteric heat of adsorption refers to the heat released by the increase in system entropy when the kinetic energy of gas molecules is converted into potential energy after adsorption when the adsorption amount is constant. It reflects the average adsorption and release heat of the unit molecule and indicates the strength and stability of the adsorption reaction [44,45,46].

Isosteric heat was calculated by Formula (7):(7)$$Q=\left[RT_{1}T_{2}\ln\left(P_{2}/P_{1}\right)\right]/\left[1000\left(T_{2}-T_{1}\right)\right]$$
where *Q* is isosteric heat, kJ/mol; *P*_1_ and *P*_2_ are the equilibrium pressure, MPa; *T*_1_ and *T*_2_ are the thermodynamic temperatures, K; R is a constant, 8.314 J/(mol·K).

As shown in Figure 7, under the current formation temperature (369.97 K) and formation pressure (28.07 MPa), the adsorption heat of methane molecules in the pores of different minerals is significantly different, but lower than the lower limit of chemical adsorption heat (84 kJ/mol), belonging to physical adsorption. Among these five minerals, adsorption heat of kerogen I and smectite are largest, respectively, 9.598 kJ/mol and 9.439 kJ/mol. Adsorption heat of chlorite is less than kerogen I and smectite, 6.255 kJ/mol and that of illite is 4.289 kJ/mol and adsorption heat of quartz is the least, only 2.096 kJ/mol (Figure 7).

Kerogen shows the strongest adsorption and exothermic reaction of a unit molecule and has the most kinetic energy from conversion, so the adsorption layer is hardly desorbed. The adsorption of methane molecules by smectite is strongest among clay minerals, and the amount of adsorption heat is close to kerogen I.

### 2.5. Evaluation of Adsorption Capacity

The data of four evaluation parameters of the above five shale matrix minerals are standardized, and the radar map of adsorption capacity evaluation is shown in Figure 8.

Absorption captivity in descending order is kerogen I > smectite > chlorite > illite > quartz.

The hydrogen bond, which is a combination of a hydrogen atom from methane and an oxygen atom from the surface of the pore wall, is the major factor controlling the adsorption effect of the pore wall in the nano pore [47]. For inorganic clay minerals, the modeling process involves using arc surfaces of cylindrical pores to dig matrix molecules, so opportunities of all the atoms and bonds to be exposed are uniform. On this basis, according to the chemical formulas of smectite (Al_2_[Si_4_O_10_](OH)_2_), illite (KAl_2_[(OH)_2_AlSi_3_O_10_]) and chlorite (Mg_4_FeAl[AlSi_3_O_10_](OH)_8_), the chemical proportions of oxygen atoms are calculated, which could measure the chance of its exposure on the pore wall.

The calculation results are as follows.

Smectite (0.5333) > chlorite (0.4915) > illite (0.4824)

This trend is consistent with the adsorption capacity of the three clay minerals, and it is concluded that the exposure probability of oxygen atoms on the pore wall surface is the main controlling factor of the adsorption capacity of clay minerals.

In conclusion, the different minerals have significant effects on the adsorption of methane molecules in shale. With the same pore form, different minerals expose different atoms and chemical bonds in the cylindrical pores, resulting in strong or weak adsorption capacity. For methane adsorption, under formation temperature of 369.97 K and formation pressure of 28.07 MPa, in the cylindrical pore with a diameter of 6.0 nm, the proportion of excess adsorption amount of kerogen I is 13.418%. In addition, the adsorption potential, the maximum adsorption, the degree of constraining adsorbate molecules and physical adsorption heat are better than other inorganic minerals. The stronger adsorption potential makes the methane molecules absorb on the surface of the pore wall firmly and intensively. The distance between the molecules is small, and a thick adsorption layer of high density is formed (7.75 Å). Among the clay minerals, smectite has the strongest adsorption capacity, followed by chlorite and illite. Because the adsorption force of the pore wall is weaker than the interaction repulsion force between adsorbents of a small distance, layered adsorption occurs (thickness of single layer is 4.22~4.92 Å). Since the hydrogen bond is stronger than the Van Der Waals force, the adsorption capacity of inorganic clay minerals is positively correlated with the formation opportunity of the hydrogen bond. Clay mineral crystals that can expose more oxygen atoms are more likely to form a hydrogen bond and bound methane. Quartz has no apparent adsorption potential. In addition, the saturated gas content calculated by the simulation experiment is similar to the measured data of the physical adsorption experiment, which proves that the Monte Carlo isothermal adsorption simulation of this research model is close to the real geological conditions and has evaluation significance.

## 3. Conclusions

(1) The relative density model shows that kerogen I has the strongest adsorption capacity and is characterized by high-density thick layer adsorption because adsorption potential overcomes the molecular repulsion. For clay minerals, adsorption force of the pore wall is less than the interaction repulsion force between adsorbents within a short distance, so 2~4 layers of narrow layer and low-density adsorption are shown. Its adsorption capacity adsorption is inversely proportional to the thickness of the single layer, and the adsorption capacity is positively correlated with the opportunity of hydrogen bonds formed by exposing oxygen atoms. In addition, quartz has no obvious adsorption potential to methane molecules. The adsorption capacity order from large to small is kerogen I > smectite > chlorite > illite > quartz.

(2) The diffusion coefficient model shows that diffusion coefficient of methane in different minerals in descending order is kerogen I < smectite < chlorite < illite < quartz. The diffusion coefficient of methane molecules in kerogen I is the smallest, and many methane molecules are bound on the sides of the pore walls and have a low degree of freedom. Because there are many hydrogen bonds between Kerogen I and methane molecules and high adsorption potential, most of the kinetic energy of methane molecules can be converted to the Van Der Waals potential energy and binding energy hydrogen bonds and methane molecules can be attached near the pore walls. Due to the combination of oxygen atoms and hydrogen bonds, clay minerals have a certain adsorption potential. The diffusion coefficient is lower than kerogen but higher than quartz. However, methane molecules in quartz have the greatest diffusion coefficient and the weakest adsorption.

(3) The saturated adsorption amount model suggests that kerogen I has the strongest adsorption capacity, and the saturated adsorption amount is up to 3.060 cm^3^/g, followed by smectite, chlorite and illite. Additionally, quartz has the weakest adsorption capacity, only 2.471 cm^3^/g. Furthermore, the mineral composition can be obtained through XRD experiment, and it is assumed that the simulated saturated adsorption amounts of the feldspar, pyrite and other minerals are similar to quartz. Then, the saturated adsorption amount can be calculated by the relevant formula.

(4) The isosteric heat of adsorption model shows that the adsorption heat of methane molecules in the pores of different minerals is greatly different, but lower than the lower limit of chemical adsorption heat, belonging to physical adsorption. Among these five minerals, adsorption heat of kerogen I and smectite are the largest, respectively, 9.598 kJ/mol and 9.439 kJ/mol. Adsorption heat of chlorite is less than kerogen I and smectite, 6.255 kJ/mol and that of illite is 4.289 kJ/mol and adsorption heat of quartz is the least, at only 2.096 kJ/mol.

## Figures and Tables

**Figure 1 nanomaterials-12-04037-f001:**
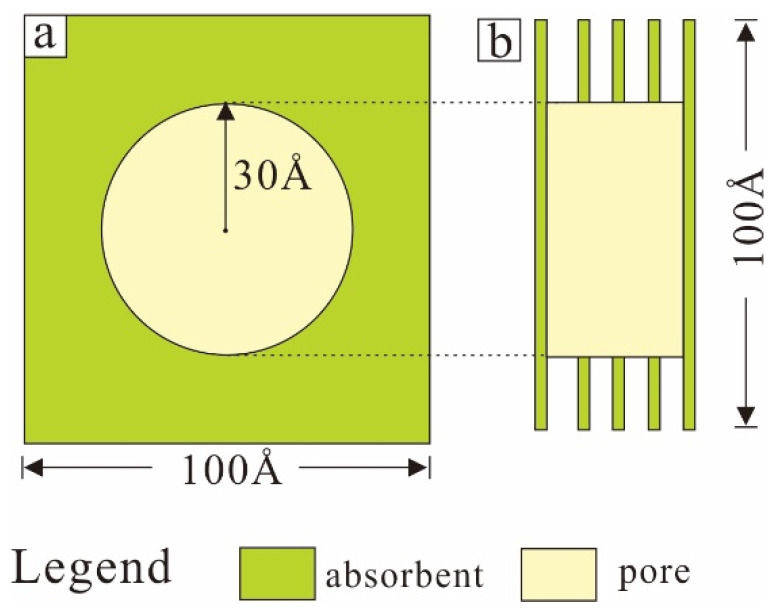
Schematic diagram of 60Å pore model. (**a**). XY direction; (**b**). Z direction.

**Figure 2 nanomaterials-12-04037-f002:**
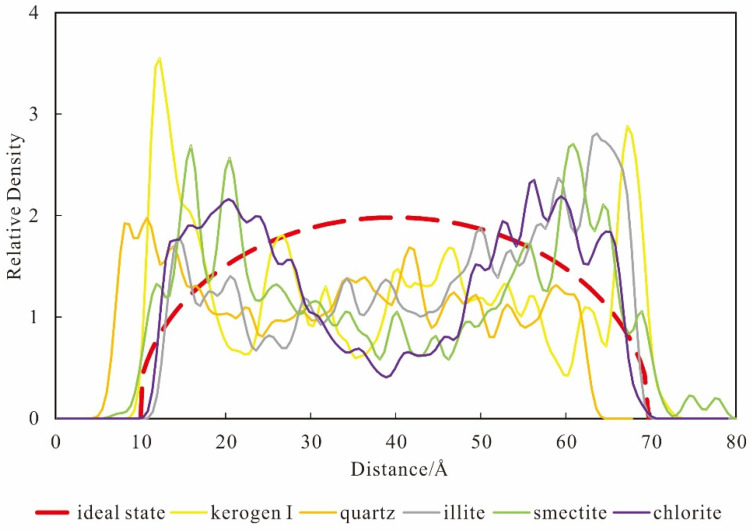
Diagram of relative density of pores in shale with different mineral matrix.

**Figure 3 nanomaterials-12-04037-f003:**
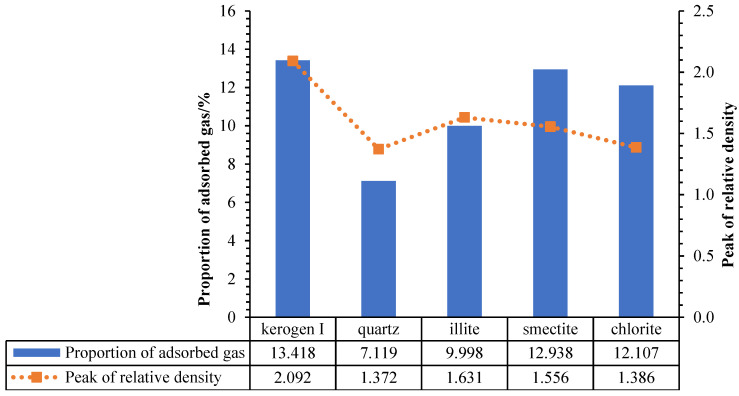
The proportion of excess adsorption amount and peak of relative density in the pores of different minerals. Kerogen I (13.418%) > smectite (12.938%) > chlorite (12.107%) > illite (9.998%) > quartz (7.119%).

**Figure 4 nanomaterials-12-04037-f004:**
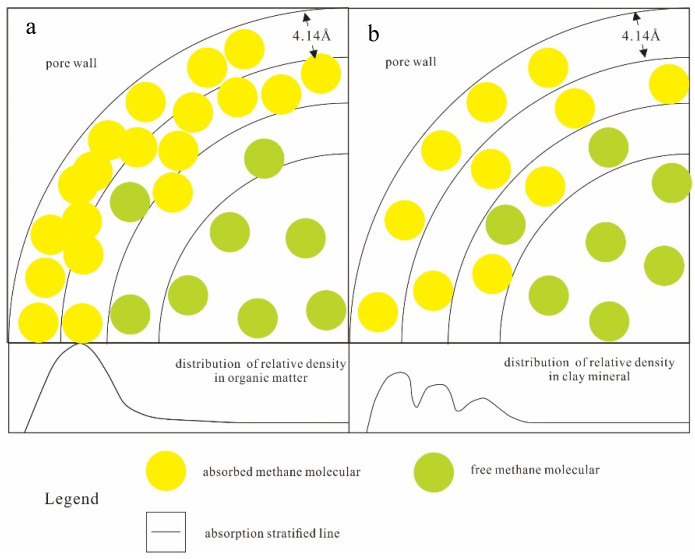
The distribution of methane molecules in shale nanoparticles. (**a**). in organic matter; (**b**). in clay mineral).

**Figure 5 nanomaterials-12-04037-f005:**
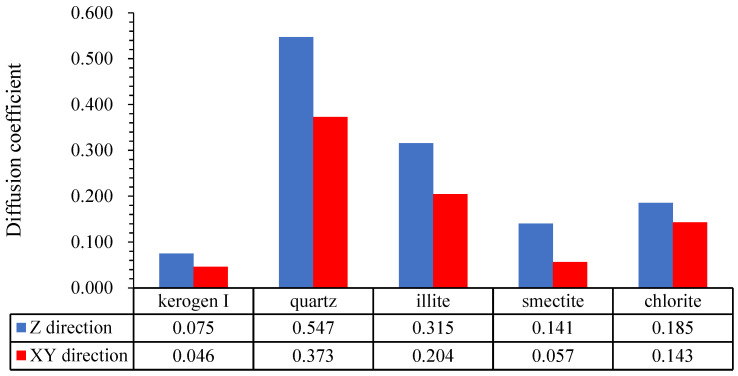
Diffusion coefficient of different minerals.

**Figure 6 nanomaterials-12-04037-f006:**
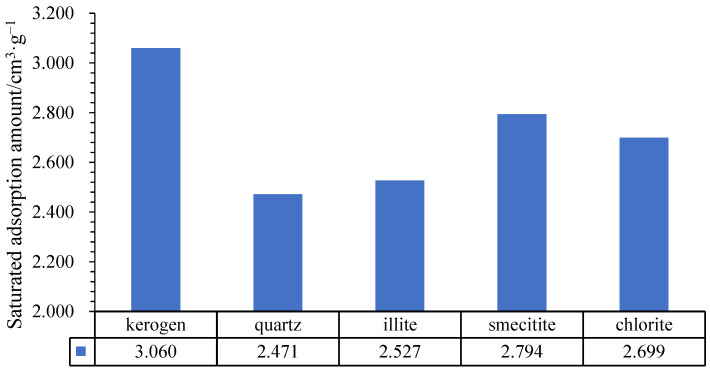
Different saturated adsorption amount of different minerals.

**Figure 7 nanomaterials-12-04037-f007:**
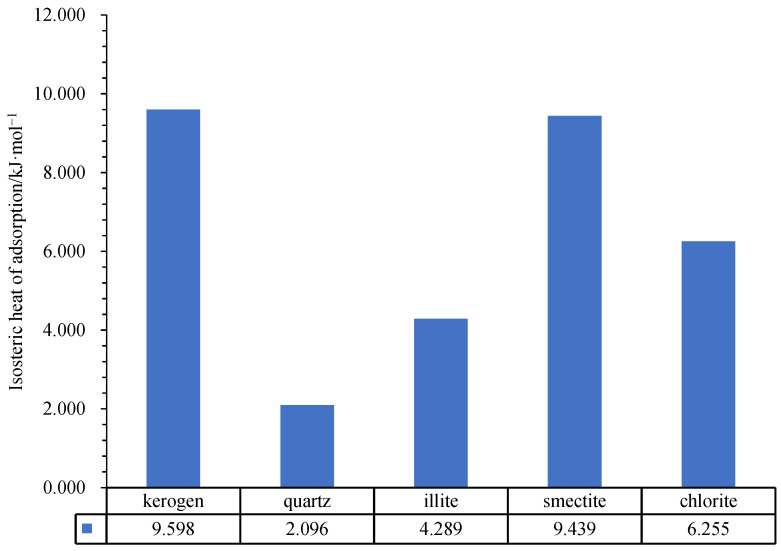
Isosteric heat of adsorption of different minerals.

**Figure 8 nanomaterials-12-04037-f008:**
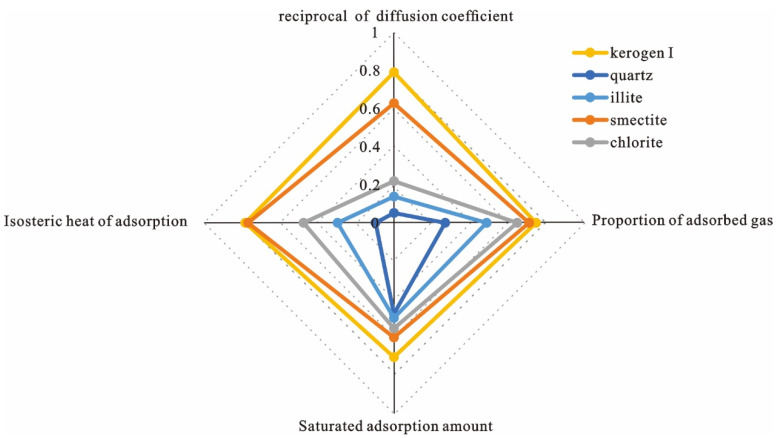
Comprehensive evaluation of adsorption capacity of different mineral matrix.

**Table 1 nanomaterials-12-04037-t001:** Geometry parameters of pore models.

Matrix Type	Pore Radius/Å	Length of OA/Å	Length of OB/Å	C plane angle/°	Length of OA′/Å	Length of OB′/Å
Kerogen I	30	84.000	80.000	90.00	84.000	80.000
Quartz	30	78.560	78.560	120.0	68.035	68.035
Illite	30	83.562	81.165	90.00	83.562	81.165
Smectite	30	82.880	80.820	90.00	82.880	80.820
Chlorite	30	79.905	83.097	89.97	79.905	83.097

**Table 2 nanomaterials-12-04037-t002:** Free Volume of pore model for several minerals.

Material	*V*_free_/Å^3^
Kerogen I	79,176.51
Quartz	84,321.07
Illite	51,395.04
Smectite	73,133.28
Chlorite	68,624.09

## Data Availability

Some of the data are contained in a published source cited in the references. All the data in this article is accessible to the readers.

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
