# Peer review of "Molecular Simulation of Methane Adsorption Capacity of Matrix Components of Shale"

_nanomaterials, 2022, doi:10.3390/nano12224037_

Round 1
Reviewer 1 Report
1. What is the novely of this work ? More details needs to be added in introduction ?
2. More Details on the determination of relative density and adsorbed gas amount from the model will need to discussed.
3.How did authors conclude higher adsorption forces in kerogen relate to smectite ?
4. Was there any past work done on the diffusion coefficient and adsorption capacity of these selected materials ? Appropriate references needs to be added and discussed.
Author Response
- What is the novely of this work ? More details needs to be added in introduction?
Reply:Thank for the reviewer’s suggestion. In general, there are 3 main problems in previous studies: (1) lack of quantitative shale gas adsorption capacity evaluation model based on molecular simulation; (2) lack of adsorption simulation under the conditions of temperature and pressure in situ; (3) not perfect shale nanopore molecular model. Innovation aims at improving these three points: (1) establish a shale quantitative evaluation model of the micro/nano pore adsorption capacity; (2) adsorption simulated by Monte Carlo and Molecular Dynamics Simulation analysis of the factors affecting the absorption ability of shale micro/nano pore and its control mechanism; (3) based on the analysis, choose the shorter edges of a perpendicular to the substrate as pore shape, cylindrical hole rather than simple slit pore or carbon nanotubes. The details are as follows.
Currently, researches on adsorbed gas are generally based on macroscopic physical simulation experiments such as methane isothermal adsorption experiment. The applied interpretation models are all based on monolayer adsorption of adsorbent, which can only measure the adsorption capacity by saturated adsorbed gas, but cannot quantitatively evaluate the adsorption capacity of shale gas. Traditional normal temperature and pressure models, such as Langmuir model, are used to explain the layered adsorption of supercritical methane under high temperature and pressure, lacking discussion of adsorption mechanism under underground burial conditions. Although some scholars have used molecular simulation method to study the adsorption performance of shale reservoir to shale gas, the pore models established are mostly simple slit pores or carbon nanopores, which have single structure and single molecular properties on the exposed matrix surface (Skoulidas et al.,2002; Cooper et al., 2003; Majumder et al., 2011; Wu et al., 2015). The modeling of micro and nano pores is not closely combined with the actual geological conditions. Therefore, based on the actual geological conditions (formation temperature, formation pressure, mineral composition, pore structure), cylindrical pores perpendicular to the short side of the matrix were selected as the pore form in this paper, which can not only fit the circular pore shape well in the XY plane, but also simulate the slit shape in the Z direction of the vertical plane. Molecular Dynamics Simulation and Monte Carlo Simulation are used for adsorption simulation experiment in nanometer pores of different matrix compositions. The distribution characteristic of supercritical adsorption of methane in shale is determined, and occurrence state models and the quantitative evaluation models are built. Using the Polanyi adsorption potential theory(Dubinin, 1960) to explain the influence of different adsorption conditions on the adsorption content and distribution of methane in nanometer pores, which is conducive to the selection of favorable blocks and advantageous strata of shale gas reservoirs and the improvement of exploitation efficiency.
- More Details on the determination of relative density and adsorbed gas amount from the model will need to discussed.
Reply:Thanks for your comment. More details on the determination of relative density and adsorbed gas amount were added.
The relative density is a dimensionless quantity. If the relative condensation is a value of 2 and it means that there are twice as many target molecules in the unit plane as in the case of uniform distribution. Since the relative density degree refers to the molecular distribution density of the whole system, The relative density of the real uniform distribution should be obtained through the conversion of formula (1):
……… …… (1)
Among, Cavg refers to the average relative density under uniform distribution; Vall refers to the total volume of system, Å3; Vpore refers to the volume of the pore space, Å3.
Saturated adsorption amount reflects the upper limit of adsorption capacity of corresponding pores (Perez and Devegowda, 2017; Prusty and Padmanablhan, 2019). By transforming the Langmuir equation, the number of adsorbed molecules and the intercept of the linear relationship between the amounts of adsorbed molecules divided by the fugacity are calculated. Then, the number of saturated adsorbed molecules in the pores of corresponding matrix can be calculated, and the concentration of gas content can be converted. Finally, the saturated adsorption amounts are obtained by formula(2).
(2)
Among, A refers to the number of adsorbed molecules in the system; B refers to the adsorption amounts, cm3/g; ρ refers to sample density, g/m3; φrefers to the average porosity of the sample, %; NA refers Avogadro's number; Vm refers the molar volume of gas, which is 22.4L in the ideal state. Vfree refers to the free volume in the pore, Å3, which can be calculated by the modeling tool Atom Volumns & Surfaces, as shown in Table 1.
Table 1 Free Volume of pore model for several minerals
|
Material |
Vfree /â„«³ |
|
Kerogen I |
79176.51 |
|
Quartz |
84321.07 |
|
Illite |
51395.04 |
|
Smectite |
73133.28 |
|
Chlorite |
68624.09 |
3.How did authors conclude higher adsorption forces in kerogen relate to smectite ?
Reply: Thank the reviewer for this advice. Here is a misunderstanding in our writing. We believe that kerogen has the strongest adsorption capacity, and smectite has the strongest adsorption capacity of inorganic minerals, which is close to kerogen. It is not that the adsorption capacity of kerogen is related to smectite. We have revised the paper for possible misunderstandings.
- Was there any past work done on the diffusion coefficient and adsorption capacity of these selected materials? Appropriate references needs to be added and discussed.
Reply: Thank the reviewer for this comment. For these selected materials (kerogen, smectite, chlorite, illite and quartz), single mineral methane isothermal adsorption experiments have been carried out, and the experimental results show that the adsorption capacity is from strong to weak: kerogen > montmorillonite > chlorite > illite > quartz (Gi et al., 2012; Xiong et al., 2022), which is consistent with the simulation results. Previous studies on methane diffusion of single mineral for these materials show that under the same temperature and pressure, the diffusion coefficient of methane is quartz > illite > chlorite > montmorillonite > kerogen (Li et al., 2018), and the simulation results, which is consistent with the simulation results.
Xiong, J.; Lin, H.; Li, Y.; et al. The desorption laws of different minerals in organic-rich shale[J]. Acta Petrolei Sinica, 2022, 43(7):989-997.
Ji, L.; Qiu, J.; Zhang, T.; et al. Experiments on methane adsorption of common clay minerals in shale[J]. Earth Science, 2012,37(5):1043-1050.
Li, J. Molecular dynamics simulation and mesoscopic flow research on shale reservoir[D]. Southwest Petroleum University, 2018.

Reviewer 2 Report
The manuscript entitled "Molecular simulation of methane adsorption capacity of matrix components of shale" by Xiaoxue Liu and coworkers reports a series of simulations (Montecarlo or Molecular Dynamics) of methane absorption onto some shale mineral matrixes. The subject may be interesting for solid-state researchers, but before publication I would recommend fixing several points (major revisions). First, I would recommend a thorough english language check. the writing is not always fluid
1) My main concern is that the driving force for absorption is the hydrogen bond interaction between the mineral and methane though C-H...... X (X=O,N,F). Though C-H....O hydrogen bond is no longer a tabu in crystallography, since a lot of examples have been described in proteins and crystal engineering, I am doubtful in this case, since methane is one of the least acidic compounds (pka around 50), even in case of very polar acceptor fragments (Si-O based). How did you quantify the interaction in your trajectories? Can you share some microscopic aggregates to justify your hypothesis? Dispersion interactions may be important since they involve every atom, think of Velcro tape.
2) Please explain how the methane diameter was calculated by convolution of van der Waals
3) 50 ps trajectories are really too short for modern simulations, please increase the length to gain longer statistics
4) What is the "real" density? Is there a fake one?
5) Diffusion coefficients can be calculated with MSD analyses in simulations (or with Green-Kubo relatioms), not RMSD, that is used to compare starting and final point geometries
6) You did not give information about the simulation cells (caps?)
7) How was the isosteric heat calculated? Can you wright the formulae?
Some errors:
van der Waal -> Waals (several times)
"under the underground high temperature state again."
"when the absorbed methane molecular is "
Strange use of while at the beginning of the sentences:
"while....so" and similar
Author Response
Please see the attachment
The manuscript entitled "Molecular simulation of methane adsorption capacity of matrix components of shale" by Xiaoxue Liu and coworkers reports a series of simulations (Montecarlo or Molecular Dynamics) of methane absorption onto some shale mineral matrixes. The subject may be interesting for solid-state researchers, but before publication I would recommend fixing several points (major revisions). First, I would recommend a thorough english language check. the writing is not always fluid
Reply: Thank you for your comment. We have checked the whole text and made corrections.
1) My main concern is that the driving force for absorption is the hydrogen bond interaction between the mineral and methane though C-H...... X (X=O,N,F). Though C-H....O hydrogen bond is no longer a tabu in crystallography, since a lot of examples have been described in proteins and crystal engineering, I am doubtful in this case, since methane is one of the least acidic compounds (pka around 50), even in case of very polar acceptor fragments (Si-O based). How did you quantify the interaction in your trajectories? Can you share some microscopic aggregates to justify your hypothesis? Dispersion interactions may be important since they involve every atom, think of Velcro tape.
Reply: Thank you for your advice. Carbon atoms in kerogen, potassium ions on the surface of illite and hydrogen atoms were placed in the same simulation system, and the binding energy between molecules or atoms changing with the distance was calculated. The results show that there are oxygen atoms on both the surface of illite and quartz, oxygen atom has the strongest force on methane, followed by potassium ion on illite surface. Carbon atom on kerogen surface has the least interaction with methane, and hydrogen atom on quartz surface has zero interaction with methane (Xu et al., 2020). Therefore, we mainly discuss the exposure and bonding of oxygen atoms into hydrogen bonds in this paper. Thank you very much for your suggestion here. Our future work can focus on C-H.... O hydrogen bond for more research.
Xu, C.; Xue, H.; Li, B.; et al. Microscopic Adsorption Mechanism Difference in the Mineral Pore of Shale Gas Reservoir[J]. Special oil and Gas Reservoirs, 2020,27(4): 79-84.
2) Please explain how the methane diameter was calculated by convolution of van der Waals.
Reply: Thank you for your suggestion. The methane diameter (4.14 Å) is based on previous research results, and references (Wang et al., 2017; Gao et al., 2020; Wang et al., 2020) have been added in relevant positions.
Wang S, Pan J, Ju Y, et al. The super-micropores in macromolecular structure of tectonically deformed coal using high-resolution transmission electron microscopy[J]. Journal of Nanoscience and Nanotechnology, 2017, 17(9): 6982-6990.
Gao Z, Ma D, Chen Y, et al. Study for the effect of temperature on methane desorption based on thermodynamics and kinetics[J]. ACS omega, 2020, 6(1): 702-714.
Wang H, Yue G, Yue J, et al. Analysis of the mechanism of temperature influencing methane adsorption in coal from perspective of adsorbed layer thickness theory[J]. Arabian Journal of Geosciences, 2020, 13(1): 1-15.
3) 50 ps trajectories are really too short for modern simulations, please increase the length to gain longer statistics
Reply: Thank the reviewer for the suggestion. We increased the length to 106 steps , a total time of 1000 ps.
4) What is the "real" density? Is there a fake one?
Reply: We thank the reviewer for this commentary. Here is our mistake. What we really want to express is "relative density", which has been corrected in the corresponding places in the paper.
5) Diffusion coefficients can be calculated with MSD analyses in simulations (or with Green-Kubo relatioms), not RMSD, that is used to compare starting and final point geometries
Reply: Thank the reviewer for this advice. We used MSD analysis. The corresponding details have been added. According to Einstein equation, the diffusion coefficient of CH4 in the pores is calculated based on Mean Square Displacement of matter per unit time. This diffusion coefficient can be derived through non-equilibrium statistical thermodynamics, as shown in formula (1).
(1)
Among, D refers to diffusion coefficient of the adsorbent molecule; t refers to the simulation time, ps; N refers to the total number of adsorbent molecules; Nt refers to the statistical average number of molecular dynamics steps; ri(t) refers to the true displacement of the i-th particle's center of mass at t time.
Mean square displacement can be calculated by formula (2).
(2)
Thus, the linear slope (Kmsd) of mean square displacement (MSD) is expressed as the formula (3).
(3)
By combination of formula (1) and (3), the slope of the mean square displacement (MSD) in three-dimensional space is 6 times that of the diffusion coefficient. In other words, if the slope of the linear relation between the mean square displacement and the simulation time is taken as 1/6, the diffusion coefficient of methane molecules in the pores corresponding to different minerals can be obtained.
6) You did not give information about the simulation cells (caps?)
Reply: Thank you for your comment. We have added the relevant information in “2.1 Parameters of simulation”. The sorbent single crystal cells were obtained from ICSD2009 crystal structure database. The chemical formula of kerogen I is (C251H385O13N7S3), smectite is Al2[Si4O10](OH)2, chlorite is Mg4FeAl[AlSi3O10](OH)8, illite is KAl2[(OH)2AlSi3O10] and quartz is SiO2 (Gualtieri, 2000; Viani et al., 2006; Zanazzi et al., 2007; Ikuta et al., 2007; Ungerer et al., 2015). For each inorganic mineral and kerogen cell in three spatial directions (the origin is O, and the three sides are OA, OB, OC in turn), the matrix with the size of 80 Å × 80 Å × 30 Å was made by the supercell expansion tool (Table 1). Since the C plane angle of chlorite and quartz single crystal cell is not a right angle, it is necessary to convert the side length when calculating the projection tetragonal system.
Table 1 Geometry parameters of pore models
|
Mineral |
Pore Radius/Å |
Length of OA/Å |
Length of OB/Å |
C plane angle/° |
Length of OA'/Å |
Length of OB'/Å |
|
Kerogen I |
30 |
84.000 |
80.000 |
90.00 |
84.000 |
80.000 |
|
Quartz |
30 |
78.560 |
78.560 |
120.0 |
68.035 |
68.035 |
|
Illite |
30 |
83.562 |
81.165 |
90.00 |
83.562 |
81.165 |
|
Smectite |
30 |
82.880 |
80.820 |
90.00 |
82.880 |
80.820 |
|
Chlorite |
30 |
79.905 |
83.097 |
89.97 |
79.905 |
83.097 |
7) How was the isosteric heat calculated? Can you wright the formulae?
Reply: Thank you for your suggestion. Isosteric heat was calculated by formula (4):
(4)
Among, Q is isosteric heat, kJ/mol; P1 and P2 are the equilibrium pressure, MPa; T1 and T2 are the thermodynamic temperatures, K; R is a constant, 8.314 J/(mol·K).
Some errors:
van der Waal -> Waals (several times)
"under the underground high temperature state again."
"when the absorbed methane molecular is "
Strange use of while at the beginning of the sentences:
"while....so" and similar
Reply: Thanks for the advice. These errors have been corrected in appropriate position.

Round 2
Reviewer 2 Report
The authors replied successfully to most of my objections, and the manuscript was greatly improved. Still, the first issue is not completely answered. I understood that oxygen atoms have the most important effect on methane, but what I still don't see is how methane can "donate" electron density through hydrogen bonding. How can C-H be significantly polarized, without any polar substituents attached to the carbon? You are saying that oxygen atoms from kerogen or quartz interact with H-C of methane. Can you show this in a picture showing the hydrogen bond triad O......H-C in a trajectory snapshot? What are the distances between O and H and how much does C-H distance increase consequently?
